# Interprofessional Collaboration in Obstetric and Midwifery Care—Multigroup Comparison of Midwives’ and Physicians’ Perspective

**DOI:** 10.3390/healthcare13151798

**Published:** 2025-07-24

**Authors:** Anja Alexandra Schulz, Markus Antonius Wirtz

**Affiliations:** Research Methods in Health Sciences, University of Education Freiburg, Kunzenweg 21, 79117 Freiburg, Germany; markus.wirtz@ph-freiburg.de

**Keywords:** interprofessional collaboration, equitable communication, midwifery care, obstetric care, multi-perspective, midwives, physicians, woman-centered care, multigroup confirmatory factor analysis, test fairness

## Abstract

**Background**: Interprofessional collaboration (IPC) is considered fundamental for integrated, high-quality woman-centered care. This study analyzes concordance/differences in the perspectives of midwives and physicians on IPC and Equitable Communication (EC) in prenatal/postpartum (PPC) and birth care (BC). **Methods**: The short form of the ICS Scale (ICS-R with eight items) adapted for the midwifery context, and the EC scale (three items) were completed by 293 midwives and 215 physicians in Germany. Profession- and the setting-specific differences were analyzed using *t*-tests and ANOVA with repeated measurements. Confirmatory factor analysis with nested model comparisons test the fairness of the scales. **Results**: Midwives’ ratings of all IPC aspects were systematically lower than physicians’ in both care settings (variance component professional group: *η*^2^*_p_* = 0.227/ 0.318), esp. for EC (*d* = 1.22–1.41). Both groups rated EC higher in BC. The setting effect was less pronounced among physicians for the ICS-R items than among midwives. Violations of test fairness reveal validity deficiencies when using the aggregated EC sum score for group comparisons. **Conclusions**: Fundamental professional differences were found in the IPC assessment between physicians and midwives. The results enhance the understanding of IPC dynamics and provide starting points for action to leverage IPC’s potential for woman-centered care.

## 1. Introduction

The provision of care for mothers and infants throughout the entire care pathway takes place in a multidisciplinary care setting. All professionals involved pursue a common goal: Ensuring high-quality, safe, and woman-centered care [1]. Due to the increasing complexity and global trends in the healthcare system, interprofessional collaboration (IPC) is recognized as a potential means to optimize integrated care processes (e.g., efficient communication structures), to reduce costs (e.g., elimination of redundant structures), and to ensure the achievement of maternal and neonatal health outcomes (e.g., reduction of medical errors) [2,3,4,5,6]. IPC is defined as a continuous process in which midwives and physicians pursue a shared holistic approach to care based on a trusting working relationship and jointly negotiated rules and structures in order to promote maternal and child health and well-being [1,7,8,9]. IPC requires (i) establishing a climate of shared values by raising awareness of the profession-specific values and norms (perspective adoption) and accepting individual profession-specific differences (values/ethics) [10]; (ii) recognizing the importance of individual members in achieving shared goals, as well as clear and open communication to define roles and responsibilities [11]; and a respectful, cooperative, and informative communication culture (interprofessional communication) [12]. This form of collaboration aims to create and foster a (shared) professional identity in which one’s own “professional” boundaries are transcended, and profession-specific stereotypes (e.g., decision-making is the responsibility of medical staff) and historically shaped models are dismantled [7,13,14].

### 1.1. Potentials, Challenges, and Requirements of Successful IPC in Women-Centered Care 

The examination of the impact of IPC on maternal and newborn health outcomes proves to be complex due to inconsistent operationalization and varying construct conceptions [3,15,16]. Types and practices of IPC examined vary from simple information transfer to joint decision-making and action processes [17]. Further, IPC is often analyzed as a sub-aspect of a multimodal intervention or is used as a secondary outcome in studies of occupational psychological processes [18]. In contrast, comprehensive evidence exists on the positive effects of IPC on healthcare professionals themselves and their working environment (esp. working relationship, joint decision-making processes, and buffering against overload) [6,19,20]. Concerns focus on data protection risks and information loss that may be associated with IPC, as well as worries about the weakening different disciplines [21]. The central challenge lies in the concrete implementation of IPC in healthcare practice [14,22]. In general, pronounced hierarchical structures (health care as a negotiated order [23]), fragmentation of care, and a lack of respect and trust, as well as unclear responsibilities, role assignments, and communication barriers represent significant obstacles to successful implementation [1,14,22].

### 1.2. Multiperspective Evaluation of IPC Within the Physician–Midwife Dyad

The professional group affiliation has to be considered as an important moderator when assessing IPC from the perspective of midwives and physicians [24,25]. Studies confirm an asymmetrical assessment between the professional groups [26]: Physicians perceive IPC with midwives and nurses more positively than vice versa [27,28]. Nurses particularly perceive communication with physicians as less open [29,30]. This is in line with a midwife-specific survey in which IPC and equitable communication (EC) with physicians were rated overall as rather critically [31]. Non-medical healthcare staff generally have a more positive attitude towards IPC and a different perception of core characteristics than medical staff [32,33]. These effects are attributed in particular to the divergent professional backgrounds and fundamentally different approaches to care and childbirth between physicians and midwives [1].

Furthermore, a midwife-specific examination clearly highlighted the relevance of the care setting in assessing IPC [31]. IPC and EC with physicians were rated significantly better during birth care (BC) than in outpatient pre- and postnatal care (PPC) settings (*SRM*: 0.579–1.405) [31]. This effect is particularly pronounced in the assessment of EC (*SRM* = 0.862–1.436). EC focuses on communication behavior that promotes group esteem and internal cohesion [31]. The dependence on the care setting is attributed to specific characteristics of the healthcare system [34]. In acute clinical care, including obstetrics, close coordination between the professionals involved is traditionally required [35]. While BC is usually provided by an interprofessional team at one location, PPC is characterized by a multi-professional and autonomous manner, resulting in less direct coordination and fewer communication paths [36].

### 1.3. Multigroup Analysis and Its Prerequisites to Examine IPC Within the Physician–Midwife Dyad

IPC is primarily assessed using self-assessments [22], which are considered valuable for capturing the reality of care processes and procedures more comprehensively [37]. The Interprofessional Collaboration Scale (ICS) developed by Kenazchuk et al. [26] was designed specifically for multigroup analyses in healthcare to assess IPC from different perspectives (physicians, nurses, and allied health professionals). The ICS is available in three languages (English, German, and Italian [38]) and provides the basis for current research on IPC within the physician–midwife dyad [28,31,39]. These studies either focus on analyzing a single perspective [31] or conduct multigroup analyses whereby testing of methodological prerequisites specific to the physician–midwife dyad, such as construct validity or measurement of invariance, is not taken into account [28,39]. Multigroup analyses based on self-assessments require instruments that ensure fair comparisons between the professional groups at scale level (test fairness) [40]. To this end, it must be ensured that both the dimensional structure, as well as the strength of the item-construct association, remain invariant, i.e., stable across professional groups. If the assumption of equality is confirmed, it can be assumed that the fairness and validity of the IPC assessments are not compromised by different profession-specific characteristic profiles [40]. The importance of this prerequisite test is confirmed by a psychometric evaluation of the ICS to capture the midwives’ perspective [31]. While previous studies have assumed the three-dimensional structure of the ICS (communication, accommodation, and isolation) to be a given, this three-dimensional structure could not be confirmed for midwives [31]. The resulting reduced ICS-R scale (two-dimensional) contains eight original ICS items and three additional indicators of EC [31]. Moreover, no studies exist that examine the homogeneity or heterogeneity of the respective professional groups in their IPC assessment. This information is crucial for valid interpretation of existing study results comparing midwives and physicians [41]. If responses within a professional group are heterogeneous, individual differences in experience, context, and perspective become more important than in a more homogeneous group. It is assumed that midwives‘ ratings may be more heterogeneous than physicians’ assessments due to the diversity of their work contexts (i.e., clinical vs. non-clinical birth care, prenatal-/postnatal care) and their different employment status.

This study addresses this research gap by conducting a multigroup analysis of IPC within the physician–midwife dyad and examining the test fairness of the reduced ICS-R instrument. The multi-perspective approach enables a deeper understanding of job-related similarities and differences in IPC perceptions. This forms the basis for the development of practical solutions to promote successful IPC in women-centered care. In detail, the study aims to extend previous findings on IPC in midwifery and obstetric care by Schulz & Wirtz [31] by including and contrasting the perspective of physicians with regard to the assessments of midwives. The objective is to investigate whether and to what extent midwives and physicians assess IPC components and indicators similarly or differently. These profession differences will be analyzed for the two care settings prenatal/postpartum (PPC) and birth (BC):

**Hypothesis** **1.**Midwives rate IPC aspects (incl. EC) lower than physicians in both care settings *(main effect: professional group).*

**Hypothesis** **2.**Differences between midwives and physicians in both care settings depend on the individual ICS and EC item content *(interaction effect: items x professional group).* Differences between midwives and physicians are particularly pronounced in items relating to direct communication aspects (lower scores for midwives).

**Hypothesis** **3.**IPC aspects (incl. EC) are generally assessed more positively across professional groups during BC than in the PPC setting *(main effect: care setting).*

Additionally, it will be examined whether the proposed measurement model (two-dimensional) is valid for both professional groups or whether the understanding of the construct or the significance of the respective ICS and EC indicators differ significantly between the groups [40].

**Hypothesis** **4.**The association of each of the 8 ICS and three EC items and the two underlying latent dimensions (IPC and EC) is identical for midwives and physicians *(testing metric invariance).*

**Hypothesis** **5.**Midwives and physicians exhibit comparable variability in their assessment of IPC and EC *(testing variance homogeneity).*

## 2. Materials and Methods

This study constitutes a follow-up to the project “Structural analysis of midwifery care in the rural district of Ortenau (Southwest Germany)”, which was approved by the Ethics Committee of the German Psychological Society (DGPs; Ref: MAW 022019). The cross-sectional study consists of two data collections: a survey of midwives from April to May 2020 (SoSciSurvey tool) and a survey of physicians from February to April 2021 (Tivian tool Unipark). No personal data was collected. Since only characteristics of the individual work situation (field of activity, professional experience, and federal state) were asked in addition to the constructs of interest (IPC), no separate ethics vote had to be obtained according to the local ethics committee. An informed consent was obtained at the beginning of the anonymous online survey, by providing all participants with comprehensive information about the aim and purpose of the study, as well as data protection and personal rights.

### 2.1. Recruitment and Participants

For both midwives and physicians, recruitment took place in two stages: (1) Ad hoc samples were drawn in all 16 federal states of Germany (primary sampling unit). A total of N = 3200 freelance midwives and midwives providing out-of-hospital antenatal care were randomly selected from the midwifery databases of the state associations. Employed midwives were recruited by contacting N = 128 hospitals (random selection of N = 2 hospitals per level of care (total 4) per federal state). Gynecologists and pediatricians were identified using the medical directory of the German Society for Ultrasound in Medicine (DEGUM) and the National Association of Statutory Health Insurance Physicians (KBV). A total of N = 1788 randomly selected physicians (N = 1342 obstetrics and gynecology; N = 446 pediatrics) were contacted. Contacts generally followed the total design survey method [42] in three stages, with a time lag of three to seven days between each step: (i) digital personal invitation letter, (ii) telephone follow-up, and (iii) digital reminder. (2) Moreover, recruitment was expanded using multipliers at the level of regional and national associations (e.g., midwives association and board of the German Association for Pediatrics and Adolescent Medicine). Eligible participants were midwives and physicians who were at least 18 years old at the time of the survey, spoke German, and were working as registered midwives or practicing physicians.

### 2.2. Instruments

The Interprofessional Collaboration Scale (ICS) assesses core elements of IPC between one or more health care professional groups (e.g., physicians, nursing, and allied health professionals) [26]. A total of N = 13 indicators assess the IPC dimensions *Communication*, *Accommodation*, and *Isolation* using a 4-point Likert scale (“1”—“strongly disagree” to “4”—“strongly agree”). The confirmatory three-dimensional structure for different caregiving dyads was confirmed in a survey using a German translated version of the ICS [38]. Composite reliability proved to be acceptable to good for the reciprocal assessment of physicians and nurses for the IPC dimension *Communication* (Mc Donalds ω _original/german version_ = 0.76–0.80/0.77), *Accommodation* (ω _original/german version_ = 0.85–0.86/0.86), and *Isolation* (ω _original/german version_ = 0.71–0.76/0.72). However, the three sub-factors exhibit strong scale intercorrelations (e.g., nurses rating IC with physicians: *r* = 0.75–0.86). Moreover, difficulties regarding model fit were reported when using the scale with allied health professionals (e.g., physiotherapists, social workers, and midwives) [26].

A psychometric analysis of the ICS in the midwifery and obstetric care setting in Germany (IPC dyad: physician–midwife; perspective: midwife) could not confirm the 3-dimensional structure of the original version [31]. After eliminating five ICS-items (ICS-01, ICS-09 to ICS-12) and adding a further sub-construct, *Equitable Communication* (EC) (3 indicators; 6-point rating scale: “1” = “strongly disagree” to “6” = strongly agree), a 2-dimensional solution achieved a satisfactory model fit (CFI  =  0.991; RMSEA  =  0.025 (95% CI = [−0.004; 0.037])). Thus, it must be assumed that the IPC measurement structure and validity of the individual ICS items differ systematically in the cooperation between midwives and physicians compared to the cooperation between physicians and nurses [31]. The 8 ICS-R and 3 EC items which were identified as structural valid for physicians–midwives cooperation were used in the present study. In line with the response range of the ICS-R items, the response categories of the EC-items were coded from ‘1’—‘strongly disagree’ to ‘4’—‘strongly agree’ (intermediate levels: ‘1.6’—“mostly disagree”, “2.20”—“rather disagree”, “2.80”—“rather agree”, and “3.40”—“mostly agree”).

Additionally, occupational characteristics and the frequency of collaboration with (i) gynaecologists and (ii) paediatricians (ordinal scale: ‘never’, ‘occasionally’, and ‘often’) as well as (iii) midwives (nominal scale: ‘yes’ and ‘no’) were surveyed.

### 2.3. Data Analysis

Cases with more than 10% missing values in the ICS-R and EC scale items were excluded before data analysis (listwise deletion). In general, missing values of more than 30% reduce the reliability of statistical analyses, as the resulting uncertainties and errors outweigh the substantial information gained [43]. Missing data of up to 5% is often considered ‘missing completely at random’ (MCAR), whereby its absence is considered to be random [43]. The more conservative cutoff of 10% was applied, which, considering a decreased sample size, ensures the robustness and reliability of the analyses. Cases with less than 10% missing values were imputed using the expectation maximization (EM) algorithm in IBM SPSS Statistics 26, which corrects for biases in case missing-at-random processes (MAR; [44,45]). To test study hypotheses 1.1–1.3 analyses of variance (ANOVA) with repeated measurements [41] were carried out separately for the two care settings, PPC and BC (within-factor 1:11 items; between factor: professional group: midwives and physicians). The interaction effect (items × professional group) is crucial to examining whether differences in the items reveal varying discrepancies between professional groups. The partial eta-square was used to interpret the effect (0.01 = small; 0.06 = medium; and 0.14 = large effect size) [41]. Furthermore, *t*-tests for independent and independent samples were performed to test pairwise mean value comparisons [46]. Cohens’*d* and standardized response mean (*SRM*) were used as effect size measures for paired mean comparisons (midwives vs. physicians; physicians PPC vs. physicians BC) and for comparisons of means between settings for paired samples (midwives PPC vs. BC; 0.2 = small; 0.5 medium; 0.8 = large effect) [47]. To prevent α-error inflation despite multiple testing, an α-error adjustment according to Bonferroni was performed for each hypothesis or statistical procedure (adjusted α = α/number of individual tests performed) [41]. All statistical analyses were performed using SPSS 29.0.

Further, the transferability of the proven 2-dimensional structural model of the ICS-R and EC scale (see [31]) was examined in the physician sample using a confirmatory factor analysis (CFA; [48]), applying the maximum likelihood (ML) estimation method in AMOS 29. The modelling was carried out separately for the two care settings BC and PPC. For this purpose, a multigroup CFA model was estimated in which the data of both professional groups (midwives vs. physicians) were analyzed in an integrated way. The multigroup model fit was tested using measures of global fit [48]. Instead of the very strict and overly sample size-dependent χ^2^-test [48], the approximate fit measures root mean square error of approximation (RMSEA), and the Tucker–Lewis index (TLI) and comparative fit index (CFI) were used as the main global fit criteria [49]. RMSEA indicates the amount of information not explained by the model (good fit: <0.05; acceptable fit < 0.08) [50]. The incremental fit measures, CFI and TLI, represent the proportion of information that can be explained by the assumed model (good fit: > 0.97; acceptable fit > 0.95) [49,51].

The analysis of test fairness in profession-specific comparisons (physicians vs. midwives) was carried out by nested model comparisons [48]. These tests were conducted by first estimating the respective parameters (variances and unstandardized loadings) separately for each group. Second, the estimation is determined as restricted between professional groups (profession-group independent estimation). The equivalence between the professional groups is tested using χ^2^ difference tests: If the model fit deteriorates significantly when the parameters (variances of latent constructs ICS-R and EC, unstandardized loadings, and inter-construct covariance) are restricted to be the same between the groups, it must be assumed that the parameters are unequal between the groups. The equality of the parameters is given if the model fit does not deteriorate [40].

The internal consistency of both scales was tested in both professional groups using McDonald’s ω (omega). In comparison to Cronbach’s α, the reliability coefficient takes into account varying unstandardized loadings as well as varying error variances (assumption: τ-congeneric measurements) [52].

## 3. Results

### 3.1. Sample Characteristics and Descriptive Statistics

A total of N = 319 midwives and N = 215 physicians completed the online questionnaire. Data sets with a percentage of missing values > 10% were excluded (N = 26 midwives; N = 0 physicians). Table 1 shows the distribution of key characteristics of occupation and employment for both professional groups. The majority of the N = 293 midwives (midw) work as freelance midwives (90.1%) and primarily offer services related to postnatal care and breastfeeding (92%). A total of 56.3% frequently cooperate with gynecologists or obstetricians. The sample of physicians (phys) consists of gynecologists or obstetricians (76.3%) and pediatricians (23.7%). Approximately half of the physician’s work in direct birth care (57.2%) and cooperate with employed midwives (56.7%). Both professional groups work mainly in urban areas (midw/phys: 65.9/95.3%) and have similar average professional experience (M(SD)_midw/phys_: 18.64 (11.96)/19.82 (9.2)).

Table 2 shows the mean values of the individual ICS-R and EC items separately for both professional groups and for the care settings prenatal/postpartum (PPC) and birth (BC) care. During PPC midwives tended to “rather disagree” with the individual ICS-R items (min/max: 2.08/2.39); in BC, the ratings ranged between “rather disagree” and “rather agree” (min/max: 2.37/2.71). Similar rating patterns between care settings were found for EC (see Table 2). In PPC, ICS-R-03 (similar ideas about treatment, M = 2.39) and EC-01 (consider as a team, M = 2.10) were rated best by midwives; in BC, the IPC aspects ICS-R-05 (cooperate with the way we organize care, M = 2.71) and EC-01 (consider as a team, M = 2.87) received the highest ratings. Physicians’ mean ratings of ICS-R aspects during PPC ranged from “rather agree” to “mostly agree” (min/max: 2.74/3.28); in BC they tended to “mostly agree” (min/max: 3.12/3.54). EC exhibited a similar response pattern between care settings (see Table 2). The highest mean rating was given by physicians in PPC for the aspects ICS-R-01 (consider our work in work planning, M = 3.28) and EC-01 (consider as a team, M = 3.01). In BC, ICS-R-04 (discuss midwives’/physicians’ issues, M = 3.54) and EC-01 (consider as a team, M = 3.62) received the highest ratings by physicians. The indicators ICS-R-06 (willing to cooperate with our practice; midw/phys: 2.37/3.12) and EC-03 (perspective adoption; midw/phys: 2.36/3.24) were rated lowest in BC in both professional groups. This consistency did not apply to PPC: Midwives agreed least with the indicators ICS-R-07 (ask for our opinion, M = 2.08) and EC-02 (eye level, M = 2.00), while physicians scored lowest on the indicators ICS-R-08 (discuss their new practices, M = 2.74) and EC-03 (perspective adoption, M = 2.84).

### 3.2. Multigroup Comparison Using the ICS-R and EC Scales in the Care Settings PPC and BC

In both care settings (PPC and BC), the mean values (i.e., item difficulties) of the individual items of the ICS-R as well as the EC scale varied significantly (PPC/BC: *F* _(df = 10, 3830/10, 4140)_ = 142.02/392.648; *p* = < 0.001/ < 0.001; see Table 3). Thus, a large proportion of the variance in the responses is explained by the item themselves (main effect: items), whereby this effect is more pronounced in BC than in PPC (*η*^2^*_p_* _(PPC/BC)_ = 0.270/0.487). Moreover, in both care settings, a significant proportion of the variance can be explained by differences between professional groups (PPC/BC: *F* _(df = 1, 383/1, 414)_ = 112.37/193.16; *p* = <0.001/<0.001; see Table 3). The professional group is a more dominant source of variance in BC than in PPC (*η*^2^*_p_* (_PPC/BC)_ = 0.227/0.318). For all 11 items, the IPC rating was significantly higher for physicians than for midwives after correction for multiple tests (see Table 2; |*d*
_PPC/BC_| = 0.61–1.44/0.80–1.15, see Appendix A). Differences are generally most pronounced for the three EC items (|*d*
_PPC/BC_| = 1.22–1.41/1.22–1.39). Only ‘consider our work in work planning’ (ICS-R-01) in PPC forms an exception, as the mean values of the professional groups differ by more than one scale point (*d* = −1.44).

However, the differences between professional groups depend on the specific content of the individual ICS-R and EC items (interaction effect: items x professional group). Thus, individual items indicate varying differences between professional groups (PPC/BC: *F* _(df = 10, 3830/10, 4140)_ = 28.04/22.43; *p* = <0.001/<0.001; see Table 3). The interaction effect proved to be moderate in PPC (*η*^2^*_p_* = 0.068) and weak in BC (*η*^2^*_p_* = 0.051).

The midwives’ responses in BC were significantly higher for seven of the eight ICS items and all three EC items after correction for multiple testing (|*SRM* _ICS/EC_|= 0.32–0.52/0.51–1.01; see Table 2). In the physician sample, three of the total eight ICS items were rated significantly better in the BC after correction for multiple tests (|*d*
_ICS/EC_| = 0.48–0.52; see Table 2). EC was rated significantly better in all three aspects by physicians working in BC than by physicians working primarily in PPC (|*d*
_EC_| = 0.58–1.09).

### 3.3. Integrated Cross-Professional Structural Analysis for the ICS-R and EC Scales

The two-dimensional measurement model, integrated modelled for both professional groups simultaneously, achieved an satisfied global model fit for both care settings, PPC (χ^2^
_(df = 86)_ = 134.09; *p* < 0.001; RMSEA = 0.038 (90%CI: [0.025; 0.050]); CFI = 0.985) and BC (χ^2^
_(df = 87)_ = 134.10; *p* = 0.001; RMSEA = 0.036 (90%CI: [0.023; 0.048]); CFI = 0.978). At local level fit, all indicators in both care settings and professional groups achieve sufficient item-construct association (min. loading: 0.592; see Table 4). Both scales proved to be highly reliable in both care settings and for both professional groups (ICS-R: McDonalds ω _PPC/BC_= 0.92–0.95/0.88–0.89; EC: ω _PPC/BC_ =0.92–0.94/0.89–0.87; see Table 2). In addition, slightly stronger item-total correlations were observed for PPC in both professional groups than in BC (see Table 2).

For BC, a significant difference in the intercorrelations of ICS-R and EC (phys/midw: *r* = 0.58/0.77; ∆χ^2^
_(df = 1)_ = 55.71, *p* < 0.001; see Table 5, RM-3)) indicates that the two constructs are perceived as more distinct in the physician sample.

### 3.4. Analysis of the Profession-Specific Test Fairness of the ICS-R and EC Scales

Step 1 included the following: For PPC, the restricted multigroup model, in which the variances of the latent dimensions ICS-R and EC were equated, showed no significant change in model fit. For PPC, neither for ICS-R nor for EC did the latent variances differ between professional groups (∆χ^2^
_(df = 2)_ = 0.37l; *p* = 0.831). Thus, the latent variances can be equated between groups in the subsequent analyses without loss of information. In contrast, equating the latent variances in BC led to a significant deterioration in the model fit (∆χ^2^
_(df = 2)_ = 19.77; *p* < 0.001). Separate modelling for the respective latent dimensions indicated that the invariance violation only exists for EC at BC (∆χ^2^
_(df = 1)_ = 19.24; *p* < 0.001). The variances of the ICS-R scale for professional groups (∆χ^2^
_(df = 1)_ = 1.84; *p* = 0.175) can be considered group invariant. The empirically determined latent variance of the EC scale is generally higher in the midwife sample than in the physician sample (SD_midw_/SD_phys_ = 2.92/1.09). That means that midwives’ judgements about EC during birth care are significantly more heterogeneous than those of physicians.

Step 2 included the following: For the PPC setting, the profession independent model, in which the loadings (RM-1) and the loadings and covariance (RM-2) are assumed to be equal across professional groups, showed no significant deterioration in data fit (RM-1/RM-2: ∆χ^2^
_(df = 11/12)_ = 12.47/15.00; *p* = 0.329/.242; see Table 5). The separate modelling for the respective latent dimensions confirmed a violation of the invariance assumption only for the dimension EC (RM-EC: ∆χ^2^
_(df = 3)_ = 8.00; *p* = 0.046; see Table 5). For BC, the group independent models (RM-1 to RM-3) achieved significantly weaker model fit when testing metric invariance at the level of factor loadings and/or covariance (RM-2: ∆χ^2^
_(df = 12)_ = 102.40; *p* < 0.001; see Table 5). In accordance with PPC, the measurement structure of the latent construct EC proved to be group specific (RM-EC: ∆χ^2^
_(df = 3)_ = 21.86; *p* < 0.001; see Table 5).

Step 3 included the following: To identify violations of group invariance at the item level, separate integrated measurement models were modelled for each factor loading across both groups. This revealed that violation of metric invariance for the EC dimension in PPC is caused by item EC-03 (perspective adoption; ∆χ^2^
_(df = 1)_ = 4.42; *p* = 0.036; see Table 4). For physicians the item ‘perspective adoption’ proved to be a more valid indicator for the latent construct EC than for midwives (λ_midw_/λ_phys_ = 0.847/.903; see Table 4).

In BC, significantly different loadings were found on the items ‘eye level’ (EC-02; λ_midw_/λ_phys_ = 0.922/0.931; ∆χ^2^
_(df = 1)_ = 4.70; *p* = 0.030) and ‘perspective adoption’ (EC-03; λ_midw_/λ_phys_ = 0.735/0.781; ∆χ^2^
_(df = 1)_ = 18.35; *p* < 0.001; see Table 4). Furthermore, measurement variance was found for the ICS-R indicators: ‘adequately discussion of women and newborn care’ (ICS-R-02) and ‘similar ideas about treatment’ (ICS-R-03). These two ICS elements are perceived by midwives as significantly more valuable for the ICS construct than by physicians (ICS-R-02/03: ∆χ^2^
_(df = 1)_ = 8.02/4.32; *p* = 0.005/0.038; see Table 4).

## 4. Discussion

This study focuses on IPC in midwifery and obstetric care and meets the recommendations of previous studies suggesting an integrated analysis of the perspectives of midwives and physicians [31]. Differences between professional perspectives point to differences in the perceived cooperation and division of tasks between the professions acting jointly in midwifery and obstetric care. These may be characteristics, causes and consequences of professionally determined self-perception and perception by the other professional group, as well as (mutual) conceptions of professional roles. Additionally, the study examined the extent to which the ICS-R and EC scales [31] used are suitable for fairly and validly assessing IPC by different professional groups.

### 4.1. Research Objective—Professional Group-Specific Differences in IPC Assessment

Midwives and physicians surveyed differ significantly in all their individual responses regarding IPC in the physician–midwife dyad. In general, midwives rate IPC aspects lower than physicians. While the mean ratings of midwives on individual items range between ‘rather disagree’ and ‘rather agree’, the ratings of physicians are more positive on average, ranging between ‘rather agree’ and ‘mostly agree’. This result is in line with existing studies, in which physicians assess IPC more positively than nursing staff, allied health professionals, and midwives [24,25,26,27,28]. Significant profession-specific differences were found in the sample for both direct birth care (BC) and pre- and postnatal care (PPC).

This strength of asymmetrical ratings of the participating midwives and physician varies depending on the content of the respective item (interaction effect of professional group and items). Differences between the professional groups are most pronounced for the three EC items. In BC, a large difference was also identified for the communication-related ICS-R-02 item ‘adequately discussion of women and newborn care’. Overall, these results confirm hypothesis 1.2, which was mainly derived from studies on collaboration between nurses and physicians [29,30]. IPC indicators relating to communication behavior are rated higher by physicians. Interprofessional communication is considered an integral factor of IPC and provides the basis for joint decision-making and ensuring coordinated care [10]. The establishment of effective communication structures, the avoidance of competitive patterns and the provision of framework conditions (context level) that allow space for interaction, and collegial exchange and feedback (e.g., supervision) are considered essential starting points for reducing discrepancies at the communicative level between midwives and physicians [1].

Furthermore, midwives rate the single items ‘consider our work in work planning’ (ICS-R-01) for BC and ‘discuss midwives’/physicians’ issues’ (ICS-R-04) for PPC markedly lower than physicians. Of all ICS-R items physicians rate these items the highest. The lower rating of the integration of midwives‘ work in physicians’ work planning during PPC could be due to the multi-professional nature of pre- and postnatal care: Both professional groups tend to work independently [53]. Hence, the development of guiding principles for cooperation is associated with considerable additional effort [1]. Accordingly, expectations regarding how much involvement of other professional groups is useful for one’s own work may vary between professional groups. With regard to the discrepancies in the discussion of clinical issues during BC, physicians may interpret the item ‘discuss clinical issues’ (ICS-R-04) primarily in the context of pathological cases. Here, responsibility and decision-making authority are primarily assigned to the medical profession. In contrast, midwives may have rated the item in the context of physiological processes, where decisions are negotiated by taking both perspectives into account [53]. In future studies, it would be useful to specify this (e.g., by using a short case study) to create a uniform basis for evaluation.

Despite systematic differences between the professional groups surveyed, a common trend can be identified: Mutual perception as a team is rated most positively by midwives and physicians in both settings compared to other EC items. Even though this assessment is significantly more pronounced among physicians, there is still a common understanding of the need for teamwork in midwifery and obstetric care. To strengthen this awareness in practice, it would be important to promote the inner conviction of healthcare professionals. The belief that the combined expertise of the care team enriches their own work and contributes to solving complex care problems strengthens the maintenance of IPC despite existing barriers [10,14].

Another practical starting point for overcoming discrepancies between professions can be found in the context of BC: Both midwives and physicians rate the aspect of ‘willing to cooperate with [our] practice’ (ICS-R-06) and ‘perspective adoption’ (EC-03) lowest. This suggests that bridging professional boundaries and the principle of synergy to achieve common care goals have not been sufficiently internalized in this sample [1]. A review that examined the factors promoting IPC based on observations of healthcare professionals concludes that professionals with a high willingness to engage in IPC are characterized by their active efforts to understand the specialist areas of other professions. They strategically plan social gatherings to promote relationship building, facilitate the adoption of perspectives, and improve the exchange of information [14]. Measures aimed at mutual education and negotiation processes are also considered useful. These strategies are intended to clarify overlaps (e.g., areas of responsibility) arising from IPC and to prevent potential conflicts [14]. These parallels are not evident for PPC. The participating midwives feel dissatisfied due to a lack of consideration of their opinions (ICS-R-07); physicians rate the ‘willingness to discuss their new practices with us’ (ICS-R-08) unsatisfactory. Both items reflect the aspect of isolation in collaborative work.

The present findings contribute to the university-wide activities initiated by the recent academization of the midwifery profession in Germany by providing valuable input for the curriculum design of new bachelor’s degree programs [54]. A stronger integration of the development of IPC competencies in the education of physicians and midwives would offer the opportunity to promote integrative working methods at an early stage and to minimize isolation processes in the long term. To support the formation of common care goals and the willingness to adopt different perspectives, it may be useful to open up study programs across disciplines and at the same time anchor interdisciplinary responsibilities at the module level [2]. It would also be beneficial to develop further training and education programs to strengthen communication skills within teams, particularly in non-hospital care, thereby maximizing the potential for synergies within interdisciplinary teams.

### 4.2. Research Objective 2—Setting-Specific Differences in IPC Assessment

The association with the care setting is particularly evident in the three EC single items in both samples examined. Midwives and physicians rate team spirit (EC-01), working at eye level (EC-02), and the ability to adopt different perspectives (EC-03) significantly better in BC than in PPC. Characteristics of the healthcare system may reinforce this setting effect, as midwives and physicians are legally obliged to form a care team in the context of acute clinical care [53]. In this context, both professions are legally equal and should consider each other’s perspectives in negotiation processes [14]. The analysis of the ICS-R items reveals a different picture: Midwives rated seven of the eight ICS-R items higher in BC. In the physician sample, the effect was marginal (three of eight ICS-R items). It should be noted that a direct comparison of the professional groups with regard to setting effects is not possible due to the different types of samples (midwives: dependent sample; physicians: independent sample). While the results in the midwife sample may highlight intra-individual differences, the results for physicians reflect structural differences between the groups (general practitioners vs. obstetricians). The lower proportion of gynecologists in private practice (18.1%) and pediatricians (23.7%) could also cause potential distortions in the results for PPC. Nevertheless, the results provide an indication that the care setting appears to be an important moderator in the midwife sample. Validation of the setting effect among physicians would require a larger sample with a balanced distribution of physicians in private practice and gynecologists working in clinical practice.

### 4.3. Research Objective 3—Test Fairness of the ICS-R and EC Assessment in Multigroup Comparisons

The two-dimensional structure of the ICS-R and EC scales developed in a sample of midwives proved to be also valid for the sample of physicians. Thus, the assessment provides a construct-valid survey instrument for operationalizing IPC in the physician–midwife dyad [31]. The ratings within the midwife sample are significantly more heterogeneous than those of physicians in BC. The observed variance of the latent construct EC is almost three times higher for midwives than for physicians (SD_midw/phys_ = 2.92/1.09). While the examined physicians arrive at similar judgments, the marked disparity in midwives’ perspectives points to significant interindividual differences in their experiences of care practice. Different working contexts for midwives in obstetrics and their employment conditions could contribute to the variability in the midwives‘ ratings. Furthermore, midwives often need to be more adaptive and flexible in their work arrangements to respond to challenges that arise in their interactions and coordination with physicians. This may contribute to greater sensitivity to communication problems only in BC. These findings should be considered to a greater extent in future surveys analyzing the midwives’ perspective. Particularly in samples with a balanced distribution of clinical and non-clinical midwives, it would be important to investigate in detail which factors cause this heterogeneity. This expanded understanding opens up the possibility of developing tailored interventions by designing specific training programs or tailored supervision programs for the relevant subgroups of midwives.

Analyzing test fairness revealed deficiencies in validity of the EC aggregated scale mean values in comparing midwives and physicians assessments (violation of the group invariance assumption). Because of higher item construct associations, the items ‘eye level’ (EC-02) and ‘perspective adoption’ (EC-03) tend to be perceived as more indicative for the EC construct by physicians than by midwives. The lack of group invariance for the indicator ‘perspective adoption’ (EC-03) is evident in both care settings. This may indicate that the traditional role model of medical dominance has not been completely overcome by current educational approaches. Physicians tend to be more involved in negotiation processes regarding the redesign of tasks within IPC teams than other professions [13,23]. The recent academisation in Germany could help to create a balance in which both professional groups equally appreciate the importance of perspective adoption, thus laying the foundation for balanced IPC [54].

The difference in the EC scale mean value cannot be attributed to true differences in the construct but tend to be distorted by differences in the weighting of the indicators [55]. We therefore recommend analyzing individual items to consider item-specific differences in the indicator–construct association [40]. In contrast, fair comparisons between professional groups are possible in PPC using the ICS-R scale mean. However, concerns about group invariance exist when using the ICS-R scale in BC: Midwives attribute greater importance to two of the eight ICS items (ICS-R-02: ‘adequately discussion of women and newborn care‘; ICS-R-03: ‘similar ideas about treatment‘) for the underlying construct than physicians. This advanced examination of methodological quality is generally required as an analytical standard for health services research in order to analyze group differences meaningfully and in accordance with measurement theory [56,57]. This increases the content-related validity and usefulness of the assessment substantially.

### 4.4. Limitations

This study is based on self-assessments by midwives and physicians, which are subject to methodological limitations. Due to the cross-sectional design, no causal conclusions can be made. Cognitive errors in judgement, such as self-serving bias, acquiescence, recall bias, and social desirability or halo effect due to positive care outcomes, could distort the response tendencies [58]. Particularly in PPC, the tendency towards extreme values or halo effects cannot be ruled out due to the higher proportion of independent work and thus a lower need and opportunity for communication [58]. This can cause floor or ceiling effects, leading to reduced variance and reduced test power. Furthermore, self-serving bias might lead to the tendency to attribute negative IPC experiences to the incorrect behavior of other team members, while positive experiences might be seen as a result of one’s own IPC skills [58]. If successful IPC processes are attributed primarily to one’s own performance, even though they are the result of the collective efforts of the entire team, existing tensions and misunderstandings between professions could increase further, and the disparity in perception of IPC between professional groups could become more pronounced. Nevertheless, self-assessments by healthcare professionals represent relevant quality assurance parameters in healthcare research, aimed at obtaining a comprehensive picture of the reality of care [37]. Both samples are ad hoc samples, which may distort the distribution of job-specific characteristics (e.g., employment status, type of work, and care level at the maternity hospital) and limit the generalisability to other, underrepresented subgroups. The lack of control measures during the recruitment process led, for example, to the participation of predominantly freelance midwives in the study; thus, clinically employed midwives tend to be underrepresented. In addition, the sample of physicians consists predominantly of specialists in gynaecology and obstetrics. The generalisability of the results is therefore limited, particularly in an international context, due to the structure of the German healthcare system [41]. To increase the success rate in recruiting clinically active midwives and physicians in the non-clinical sector, efforts should be made to strengthen motivational factors during the pre-intentional phase of study participation in line with the INTACT-RS framework [59]. To this end, individuals with particular motivation should be identified and involved as peer recruiters (e.g., senior midwives in the delivery room; members of a medical quality circle). Material incentives could also be used (e.g., monetary donations to social initiatives).

It would also have been desirable to supplement and contextualize the available results with qualitative data to deepen the significance of the quantitative data. For example, detailed information on expectations regarding the intensity of involvement of other professional groups during PPC could have enriched the interpretation of the differences found in the ICS indicators ‘consider our work in work planning’ (ICS-R-01) and ‘discuss midwives’/physicians’ issues’ (ICS-R-04). In future studies, it is recommended to recruit physician–midwife dyads or teams. Using multilevel analysis, individual (e.g., communication style) and dyadic influences (e.g., trust between partners) on the IPC process and its appraisal could be investigated [41].

## 5. Conclusions

The present study highlights fundamental profession-related differences in the assessment of IPC between midwives and physicians surveyed. The multigroup comparison provides a deeper understanding of IPC processes and dynamics in the reality of care for pregnant women, women in labor, women in the postpartum period and breastfeeding women. The multi-perspective approach is essential for understanding profession-related challenges in implementing IPC in care practice and for developing practical solutions. The results indicate a need for action regarding EC and point out the importance of tailoring future measures to the specific needs and dynamics of the professional groups involved. For example, implementing IPC protocols in the perinatal setting could be helpful in identifying successful and obstructive IPC processes and developing joint solutions during regular meetings or quality circles. Furthermore, the results on the test fairness of the ICS-R and EC scales contribute to broadening existing knowledge on the psychometric properties of assessment standards for women-centered and interprofessional care and to clarifying their relevance and applicability to the context of midwifery and obstetric care [55].

## Figures and Tables

**Table 1 healthcare-13-01798-t001:** Descriptive sample characteristics.

	Midwives (N = 293)	Physicians (N = 215)
**Professional experience (years)**	M = 18.64 SD = 11.96 mode = 18.00	M = 19.82 SD = 9.20 mode = 20.00
**Field of Specialty**		
gynecology/obstetrics	-	164 (76.3)
pediatrics	-	51 (23.7)
**Employment**		
employed	29 (9.9%)	-
freelance	264 (90.1%)	-
**Scope of activity** ^ (1) ^		
prenatal/pregnancy	213 (71.2%)	39 (18.1%)
perinatal/birth	174 (58.2%)	132 (57.2%)
postpartum	275 (92.0%)	53 (24.7%)
**Work location**		
urban areas	193 (65.9%)	205 (95.3%)
rural areas	94 (32.1%)	10 (4.7%)
**Scope of employment**		
full-time	149 (50.6%)	159 (74%)
part-time	143 (48.8)	56 (26%)
**Cooperation with paediatricians**		
frequently	101 (34.5%)	-
occasionally	176 (60.1%)	-
never	16 (5.5%)	-
**Cooperation with gynaecologists**		
frequently	165 (56.3%)	-
occasionally	118 (40.3%)	-
never	10 (3.4%)	-
**Cooperation with midwives**		
employed midwives	-	122 (56.7%)
freelance midwives	-	93 (43.3%)

Legend: ^(1)^ multiple choice possible; M = mean value; SD = standard deviation.

**Table 2 healthcare-13-01798-t002:** Descriptive statistics and measures of local item fit of the ICS-R and EC indicators in prenatal and postpartum care (PPC) as well as birth care (BC) for midwives and physicians.

IC Items		PPC N = 293 Midwives, N = 92 Physicians				BC N = 293 Midwives, N = 123 Physicians				PPC-BC	
		*M (SD)*	*d* [95% CI]	*r_it_* ^(2)^	ω	*M (SD)*	*d* [95% CI]	*r_it_* ^(2)^	ω	M (se)	*d/SRM*^(3)^ [95% CI]
Reduced Interprofessional Collaboration Scale (ICS)-R [ratings: 1 (strongly disagree)–4 (strongly agree)]											
ICS-R-01: […] are usually willing to take into account the convenience of [us] when planning their work	Midwives	2.13 (0.81)	−1.44 *** [−1.69; −1.18]	0.71	midw: 0.92 phys: 0.95	2.63 (0.80)	−0.80 *** [−1.01; −0.59]	0.56	midw: 0.88 phys: 0.89	−0.50 (0.06)	−0.52 *** [−0.64; −0.40]
	Physicians	**3.28 (0.76)**		0.83		3.27 (0.78)		0.64		0.01 (0.11)	(0.02) ^n.s.^ [−0.25; 0.29]
ICS-R-02: I feel that woman and newborn care are adequately discussed between midwives and physicians	Midwives	2.31 (0.86)	−0.83 *** [−1.07; −0.58]	0.74		2.61 (0.84)	−1.11 *** [−1.33; −0.88]	0.68		−0.30 (0.05)	−0.32 *** [−0.44; −0.21]
	Physicians	3.03 (0.91)		0.76		3.49 (0.67)		0.56		−0.46 (0.11)	(−0.58 **) ^(1)^ [−0.86; −0.31]
ICS-R-03: The physicians and midwives have similar ideas about how women and newborn should be treated	Midwives	**2.39 (0.82)**	−0.61 *** [−0.58; −0.37]	0.65		2.49 (0.81)	−0.93 *** [−1.15; −0.71;]	0.64		−0.10 (0.05)	(−0.12 *) ^(1)^ [−0.24; −0.01]
	Physicians	2.80 (0.83)		0.77		3.20 (0.64)		0.63		−0.30 (0.10)	(−0.42 **) ^(1)^ [−0.69; −0.15]
ICS-R-04: […] are willing to discuss midwives’/physicians’ issues	Midwives	2.32 (0.90)	−0.91 *** [−1.15; −0.67]	0.80		2.63 (0.84)	−1.15 *** [−1.37; −0.92]	0.69		−0.31 (0.05)	−0.38 *** [−0.49; −0.26]
	Physicians	3.15 (0.94)		00.87		**3.54 (0.66)**		0.75		−0.38 (0.11)	−0.49 *** [−0.76; −0.21]
ICS-R-05: […] cooperate with the way we organize midwifery/obstetric	Midwives	2.38 (0.83)	−0.88 *** [−1.12; −0.64]	0.75		**2.71 (0.73)**	−0.87 *** [−1.09; −0.65]	0.65		−0.33 (0.05)	−0.38 *** [−0.50; −0.26]
	Physicians	3.11 (0.82)		0.80		3.33 (0.67)		0.73		−0.22 (0.10)	(−0.29 *) ^(1)^ [−0.57; −0.02]
ICS-R-06: […] would be willing to cooperate with midwifery/physician practices	Midwives	2.13 (0.77)	−0.89 *** [−1.13; −0.64]	0.77		** 2.37 (0.75) **	−1.04 *** [−1.26; −0.81]	0.67		−0.24 (0.04)	−0.32 *** [−0.44; −0.20]
	Physicians	2.83 (0.86)		0.86		** 3.12 (0.67) **		0.69		−0.30 (0.10)	(−0.39 **) ^(1)^ [−0.66; −0.12]
ICS-R-07: […] usually asks for [our] opinion	Midwives	** 2.08 (0.94) **	−0.70 *** [−0.94; −0.46]	0.73		2.52 (0.85)	−0.85 *** [−1.07; −0.63]	0.64		−0.44 (0.05)	−0.49 *** [−0.61; −0.37]
	Physicians	2.75 (1.02)		0.79		3.21 (0.76)		0.70		−0.46 (0.12)	−0.52 *** [−0.80; −0.25]
ICS-R-08: […] are willing to discuss their new practices with us	Midwives	2.10 (0.89)	−0.73 *** [−0.97; −0.49]	0.71		2.38 (0.82)	−0.94 *** [−1.16; −0.72]	0.54		−0.28 (0.05)	−0.32 *** [−0.43; −0.20]
	Physicians	** 2.74 (0.89) **		0.81		3.12 (0.72)		0.64		−0.38 (0.11)	−0.48 *** [−0.75; −0.21]
Scale value ICS-R	Midwives	2.23 (0.68)	−1.06 *** [−1.31; −0.82]			2.54 (0.59)	−1.31 *** [−1.54; −1.08]			−0.31 (0.03)	−0.59 *** [−0.71; −0.42]
	Physicians	2.97 (0.75)				3.28 (0.52)				−0.31 (0.09)	−0.49 *** [−0.77; −0.22]
Equitable communication (EC) [ratings: 1 (strongly disagree)–4 (strongly agree)]											
EC-01: Physicians and midwives consider themselves as a team	Midwives	**2.10 (0.72)**	−1.27 *** [−1.52; −1.01]	0.83	midw: 0.92 phys: 0.94	**2.87 (0.68)**	−1.22 *** [−1.45; −1.00]	0.73	midw: 0.89 phys: 0.87	−0.77 (0.04)	−1.01 *** [−1.15; −0.87]
	Physicians	**3.01 (0.69)**		0.87		**3.62 (0.44)**		0.69		−0.61 (0.08)	−1.09 *** [−1.38; −0.80]
EC-02: Physicians and midwives encounter at eye level	Midwives	** 2.00 (0.68) **	−1.41 *** [−1.66; −1.15]	0.87		2.49 (0.77)	−1.26 *** [−1.48; −1.03]	0.82		−0.49 (0.04)	−0.65 *** [−0.78; −0.52]
	Physicians	2.96 (0.70)		0.89		3.39 (0.57)		0.82		−0.43 (0.09)	−0.68 *** [−0.95; −0.40]
EC-03: […] try to place themselves in the perspective of the other professional group	Midwives	2.04 (0.63)	−1.22 *** [−1.47; −0.97]	0.81		** 2.36 (0.63) **	−1.39 *** [−1.62; −1.16]	0.67		−0.32 (0.04)	−0.51 *** [−0.63; −0.38]
	Physicians	** 2.84 (0.74) **		0.87		** 3.24 (0.65) **		0.71		−0.40 (0.09)	−0.58 *** [−0.85; −0.30]
Scale value EC	Midwives	2.00 (0.68)	−1.41 *** [−1.66; −1.15]			2.61 (0.70)	−1.32 *** [−1.55; −1.09]			−0.61 (0.04)	−0.87 *** [−1.01; −0.74]
	Physicians	2.96 (0.70)				3.47 (0.50)				−0.50 (0.08)	−0.85 *** [−1.13; −0.57]

Legend: ^(1)^ not significant after Bonferroni correction (α_corr_ = α/13tests = 0.004); ^(2)^ Item-total-correlation of the ICS-R and EC-items, ^(3)^ subgroup midwives: *t*-test for dependent samples (effect size: standardized response means of differences (*SRM*)); subgroup physicians: *t*-test for independent samples (effect size: Cohens’*d*). CI = confidence interval; se = standard error; n.s. = not significant. Bold faced: ICS-R and EC total score statistics. *** *p* < 0.001, ** *p* < 0.01, * *p* < 0.05, bold = highest value within each occupational group separately for PPC and BC; bold and underlined: lowest value within each occupational group separately for the PPC and BC settings.

**Table 3 healthcare-13-01798-t003:** Results of the 11x2-ANOVA with repeated measures testing the within-subject effects of the ICS-R and EC items as well as the between-subject effect of professional group (midwives vs. physicians) as well as item-professional group interactions.

	PPC				BC			
	*F*	df	*p*	*η*^2^*_p_* ^(a)^	*F*	df	*p*	*η*^2^*_p_* ^(a)^
**Main effect**								
Scale items	142.016	10, 3830	<0.001	0.270	392.648	10, 4140	<0.001	0.487
Professional group	112.372	1, 383	<0.001	0.227	193.157	1, 414	<0.001	0.318
**Interaction effect**								
Items x professional group	28.043	10, 3830	<0.001	0.068	22.425	10, 4140	<0.001	0.051

Legend: PPC = prenatal/postpartum care; BC = birth care; ^(a)^ partial η ^2^ (eta) (η ^2^_*p*_ = 0.01 = small; 0.06 = medium; 0.14 = large effect size).

**Table 4 healthcare-13-01798-t004:** Measures of local model fit and parameters of item-specific invariance testing.

IC Items	PPC			BC		
	Standardized Factor Loading (Phys|Midw)	∆χ^2 (a)^	*p*	Standardized Factor Loading (Phys|Midw)	∆χ^2 (a)^	*p*
ICS-R-01: […] are usually willing to take into account the convenience of [us] when planning their work	0.848|0.745	0.261	0.610	0.684|0.592	0.529	0.467
ICS-R-02: I feel that woman and newborn care are adequately discussed between midwives and physicians	0.792|0.765	0.443	0.506	**0.592|0.725 ***	8.023	**0.005**
ICS-R-03: The physicians and midwives have similar ideas about how women and newborn should be treated	0.806|0.680	1.855	0.173	**0.663|0.703 ***	4.323	**0.038**
ICS-R-04: […] are willing to discuss midwives’/physicians’ issues	0.885|0.835	0.759	0.384	0.796|0.748	2.338	0.126
ICS-R-05: […] cooperate with the way we organize midwifery/obstetric	0.829|0.782	0.134	0.714	0.776|0.685	0.104	0.747
ICS-R-06: […] would be willing to cooperate with midwifery/physician practices	0.889|0.814	3.204	0.073	0.736|0.720	0.461	0.497
ICS-R-07: […] usually asks for [our] opinion	0.823|0.762	1.562	0.211	0.752|0.696	0.060	0.807
ICS-R-08: […] are willing to discuss their new practices with us	0.832|0.755	0.647	0.421	0.682|0.588	0.026	0.872
EC-01: Physicians and midwives consider themselves as a team	0.906|0.875	0.024	0.877	0.777|0.818	0.000	0.997
EC-02: Physicians and midwives encounter at eye level	0.941|0.946	0.077	0.782	**0.931|0.922 ***	4.697	**0.030**
EC-03: […] try to place themselves in the perspective of the other professional group	**0.903|0.847 ***	4.419	**0.** **036**	**0.781|0.735 ***	18.346	**<0.001**

Legend: ^(a)^ degrees of freedom = 1; ***** = significantly different factor loadings between professional groups.

**Table 5 healthcare-13-01798-t005:** Nested model comparisons (midwives vs. physicians) analyzing group invariance of the measurement models, ICS-R and EC.

	PPC			BC		
	∆χ^2^	df	*p*	∆χ^2^	df	*p*
**RM-1** (restricted factor loadings ICS-R and EC)	12.469 ^(1)^	11	0.329	40.976 ^(2)^	11	<0.001
**RM-2** (restricted factor loadings ICS-R and EC and covariance)	14.997 ^(1)^	12	0.242	102.398 ^(2)^	12	<0.001
**RM-3** (restricted covariance ICS-R—EC)				55.714 ^(2)^	1	<0.001
**RM-ICS-R** (restricted factor loadings ICS-R)	4.836 ^(1)^	8	0.775	14.677 ^(2)^	8	0.066
**RM-EC** (restricted factor loadings EC)	7.995 ^(1)^	3	0.046	21.860 ^(2)^	3	<0.001

Legend: ^(1)^ = fixed variance of latent constructs (ICS-R and EC) at 1.0. ^(2)^ fixed variance ICS-R at 1.0; fixed latent EC variance by means of determined latent sample characteristics (phys: 1.09; midw: 2.92). RM = restricted model.

## Data Availability

The raw data supporting the conclusion of this article will be made available by the authors upon request (anja.schulz@ph-freiburg.de).

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
