# Peer review of "Interprofessional Collaboration in Obstetric and Midwifery Care—Multigroup Comparison of Midwives’ and Physicians’ Perspective"

_healthcare, 2025, doi:10.3390/healthcare13151798_

Round 1
Reviewer 1 Report
Comments and Suggestions for Authors
The article presents a solid and methodologically rigorous investigation into interprofessional perceptions between midwives and physicians, with well-substantiated results relevant to clinical practice. The introduction and methods are clearly articulated, and the conclusions are well aligned with the data. Minor improvements are recommended regarding the articulation of practical implications and the visual presentation of results.
Suggestion:
- The introduction is well-contextualised but would benefit from a clearer articulation of the research gap – that is, more explicitly stating what has not yet been addressed in the literature and the unique contribution of this study;
- Consider ending the introduction with a concise sentence highlighting the practical and scientific value of the multigroup analysis;
- Clarify the inclusion and exclusion criteria for participants beyond the absence of personal data collection;
- Indicate whether there was any control for or analysis of potential sampling bias, as the majority of midwives were freelancers and the physicians mainly worked in clinical settings;
- Justify more explicitly why the study focused solely on self-assessments and did not incorporate peer evaluations or external observations;
- The discussion is strong, but could be further enhanced by elaborating on the practical implications – how can these findings be used to inform training policies or institutional guidelines?;
- Emphasise how the finding of greater heterogeneity among midwives might translate into tailored interventions (e.g., customised training programmes or supervision models);
- Limitations are appropriately acknowledged, but it may help to underline the issue of limited international generalisability, as the study is specific to the German context;
- Consider briefly noting the cross-sectional design as a limitation in terms of drawing causal inferences;
- The conclusion effectively summarises key implications. However, suggesting a few concrete actions for future research or practical application (e.g., development of IPC protocols specific to perinatal care) would strengthen the final message.
Author Response
Dear reviewer 1,
Please see the attached file for our point-by-point response.
Kind regards

Reviewer 2 Report
Comments and Suggestions for Authors
The article raises very interesting content, but requires modification.
The abstract needs to be shortened - especially the background.
Too many keywords
The introduction is quite interesting but definitely too long, the most important content should be selected.
Is there a hospital reference level in Germany? If so, was the division included in the research?
Author Response
Dear reviewer 2,
Please see the attached file for our point-by-point response.
Kind regards

Reviewer 3 Report
Comments and Suggestions for Authors
The authors should justify further the use of scales and consider reporting results at the item level (as they recommend in their discussion) rather than relying on aggregate scores for cross-professional comparison.
Discuss the limitation of skeweness more explicitly.
The authors should discuss in more depth how self response biases might influence differences found between groups and settings
Recommend and suggest that future research incorporate qualitative interviews or case studies to contextualize and validate the findings
The authors report that setting effects (PPC vs. BC) are more pronounced for midwives than for physicians, but also note that sample structure makes direct comparisons challenging.
The study reports that no additional ethics vote was needed, which is acceptable under local regulations, but international readers may expect formal ethics board approval
Consider expanding on why prior research has failed to capture both perspectives adequately in introduction
Benefit from the following studies
Schulz, A. A., & Wirtz, M. A. (2023). Assessment of interprofessional obstetric and midwifery care from the midwives’ perspective using the Interprofessional Collaboration Scale (ICS). Frontiers in psychology, 14, 1143110.
Romijn, A., Teunissen, P. W., de Bruijne, M. C., Wagner, C., & de Groot, C. J. (2018). Interprofessional collaboration among care professionals in obstetrical care: are perceptions aligned?. BMJ quality & safety, 27(4), 279-286.
Beier, L., Thaqi, Q., Luyben, A., Kimmich, N., & Naef, R. (2024). Predicting collaborative practice between midwives and obstetricians: A regression analysis. European Journal of Midwifery, 8, 10-18332.
Kiewan, R., Gharaibeh, M., Alnuaimi, K.,, & Aladwan, H. (2021). Attitudes of midwives and obstetricians towards midwives practiced roles in hospitals: A national study in Jordan. International Journal of Clinical Practice, 75(12), e14891.
The sample is highly skewed toward freelance midwives and OBGYNs. This limits generalizability to other practice types.
please provide rationale for 10% cutoff and discuss sensitivity to this threshold.
Avoid language suggesting causality or overgeneralizing; you are reporting associations in cross-sectional data
Tables 2 and 3 are data-rich but hard to interpret at a glance. Suggest using bold or shaded rows for key findings, and adding an “interpretation” row for major differences
Author Response
Dear reviewer 3,
Please see the attached file for our point-by-point response.
Kind regards

Round 2
Reviewer 2 Report
Comments and Suggestions for Authors
The manuscript has been revised according to comments.
Reviewer 3 Report
Comments and Suggestions for Authors
No more changes are required